# Plant DNA Methylation Responds to Nutrient Stress

**DOI:** 10.3390/genes13060992

**Published:** 2022-05-31

**Authors:** Xiaoru Fan, Lirun Peng, Yong Zhang

**Affiliations:** 1School of Chemistry and Life Science, Anshan Normal University, Anshan 114007, China; xiaorufan@mail.asnc.edu.cn; 2College of Resource and Environmental Science, Nanjing Agricultural University, Nanjing 210014, China; 2018203054@njau.edu.cn; 3Institute of Food Crops, Jiangsu Academy of Agricultural Sciences, Nanjing 210014, China

**Keywords:** DNA methylation, nutrient stress, epigenetic, plants, abiotic stress

## Abstract

Nutrient stress as abiotic stress has become one of the important factors restricting crop yield and quality. DNA methylation is an essential epigenetic modification that can effectively regulate genome stability. Exploring DNA methylation responses to nutrient stress could lay the foundation for improving plant tolerance to nutrient stress. This article summarizes the plant DNA methylation patterns, the effects of nutrient stress, such as nitrogen, phosphorus, iron, zinc and sulfur stress, on plant DNA methylation and research techniques for plant DNA methylation, etc. Our discussion provides insight for further research on epigenetics response to nutrient stress in the future.

## 1. Introduction

The adversity stresses of plants are usually divided into two types: biotic stresses, including pathogenic bacteria, insect pests and weed damage, and abiotic stresses, such as water, temperature and nutrient elements [1]. During the growth and development of crops, 17 essential nutrients are required to maintain their growth, including macronutrients, mesonutrients and micronutrients [1]. As a common abiotic stress, low or excessive levels of nutrients could cause the loss of crop yield and quality [2]. In order to cope with the effects of nutrient stress on their growth and development, plants have evolved complex mechanisms to adapt to fluctuations in the nutrients in the soil [3,4,5]. Over the past decade, the core abiotic stress signaling pathways have been gradually elucidated [1]. More studies demonstrate the vital involvement of epigenetic mechanisms in abiotic stress responses [6,7,8].

Epigenetic mechanisms play a crucial role in forming adversity-responsive memory and can be inherited by future generations [9]. Plants are often subjected to adverse environmental stress conditions due to their sessile nature. If plants have experienced stress, they can respond more quickly and have a greater chance of survival than plants that have never encountered environmental stress [10]. Resistance to stress conditions enhances plant resistance to abiotic conditions [11]. Mild exposure to stress results in a new cellular state in comparison to that of plants that have never been exposed to stress [12,13]. If the stress persists for a period, the plant can generate stressful memories [9,10,12,14]. This stress memory is usually regulated by DNA methylation, histone modifications and the accumulation of signaling proteins [15,16,17]. It has already been proven that the stress-induced changes in DNA methylation could be partially inherited by the next generation, which preferentially occurs through the female germ line [18,19]. Such heredity was considered a source of diversity which could be utilized in breeding programs [20]. Therefore, the study of plant epigenetic mechanisms has great significance for crop cultivation [21]. However, persistent stress is vital for establishing DNA methylation-dependent stress memory in plants [22]. If the progeny were not continuously stressed, the inherited epigenetic status is gradually reset [19], but how many generations are needed to establish an epigenetic memory is still unclear.

Epigenetics is the study of DNA sequence-independent changes in gene function that are mitotically and/or meiotically heritable. It plays an important role in maintaining the cellular memory of gene expression states [6,23]. Epigenetics includes chromosome configuration recombination, histone modification, DNA methylation, non-coding RNA-mediated regulation, etc. [24]. DNA methylation is one of the most thoroughly studied mechanisms in epigenetic research [25]. The dynamic regulation of DNA methylation in response to environmental changes reduces plants’ survival pressure in harsh environments and helps plants respond to stress [26,27,28,29]. The reversibility of DNA methylation could rapidly and reversibly modify plant genomic DNA, which avoids excessive gene recombination and population diversity [30]. Its heritability provides new ideas for plant breeding [31]. This review determined the mechanism of plant DNA methylation, the effects of different nutrient stresses on DNA methylation and related research techniques. Our review helps breed new plant varieties with stronger nutrient stress resistance for genome-based breeding.

## 2. Plant DNA Methylation Patterns

DNA methylation is one of the earliest discovered and most studied regulatory mechanisms in epigenetics [32]. Studies have shown that tissue-specific DNA methylation patterns in plants can be stably transmitted asexually through the complex process of regenerating intact plants from a single source tissue [33].

DNA methylation includes C5-methylcytosine (5mC), N6-methyladenine (N6-mA) and N7-methylguanine. 5mC, the fifth position of the cytosine residue, is the most widely studied DNA methylation [34]. 5mC is the fifth carbon of a cytosine residue that receives methyl from S-adenosyl-L-methionine (SAM) by the catalysis of DNA methyltransferase (DNMT) [34,35]. The 5mC in mammals is mainly found at CG sites, but in plants, it can occur in the three cytosine environments of CG, CHG and CHH (H stands for A, T or C), and they are catalyzed by different DNMTs [35,36]. Different DNMTs are involved in two DNA methylation processes in plants: DNA methylation maintenance and de novo DNA methylation (Figure 1).

Methyltransferase 1 (MET1), one of the DNMTs, mainly maintains symmetric CG site methylation, which is an ortholog of mammalian DNMT1. MET1 recognizes double-stranded DNA with hemimethylated CG and induces unmodified cytosine methylation during DNA replication [35,37]. The symmetrical CHG site methylation is primarily maintained by chromosomal methylase 3 (CMT3), which binds to the inhibitory H3K9me2 and induces unmodified CHG sites methylation [38,39]. Additionally, CHG methylation is catalyzed by CMT2 [40,41]. The suppressor of the variegation homolog protein, SUVH4, SUVH5 and SUVH6 binds to the methylated CHG site and promotes the function of CMT3/CMT2 [42,43]. During asymmetric de novo DNA methylation, CHH methylation is performed by domain-rearranged methyltransferase 2 (DRM2) or CMT2, depending on the genomic region. DRM2 causes CHH methylation through a plant-specific mechanism, the RNA-directed DNA methylation (RdDM) pathway, which depends on the 24 nt small interfering RNA (siRNA) [20,35,44].

Most of the RdDM pathway research has studied *Arabidopsis thaliana* [20,35,45]. RNA polymerase IV (Pol IV), as the template for RNA-dependent RNA polymerase 2 (RDR2), mediates the generation of double-stranded RNAs (dsRNA) [20,44]. Then, the DICER-like protein (DCL2/3/4) cleaves the dsRNAs to generate 24 nt siRNAs. siRNA is loaded onto the ARGONAUTE proteins (AGO), mainly AGO4 and AGO6, which interact with DRM2 to catalyze de novo DNA methylation [35,38,46] (Figure 1). This process is assisted by RNA-DIRECTED DNA METHYLATION 1 (RDM1), which may bind single-stranded methylated DNA [47]. At some RdDM loci, Pol II produces 24 nt siRNAs and scaffold RNAs [48]. Furthermore, at some transposons, POL II and RDR6 produce precursors of 21 nt or 22 nt siRNAs [49,50,51].

On the other hand, replacing 5mC with unmethylated cytosine is equally important in regulating gene expression as cytosine methylated. DNA demethylation includes passive demethylation and active demethylation [52,53]. In passive demethylation, 5mC loses its methyl during DNA replication, and in active demethylation, 5mC losses are catalyzed by DNA glycosylases [54]. Passive demethylation is a nuclear factor (NF) that adheres to the 5mC during DNA replication and blocks the maintenance of DNA methylation, which leads to the loss of DNA methylation in the newly synthesized strand [52,54]. Active demethylation balances the methylation level of the genome and maintains gene expression. The excision of C5-methylcytosine achieves the active demethylation of 5mC by the DNA glycosylase and then repairs the cytosine by the base excision through the base-excision repair (BER) pathway [54]. There are four DNA glycosylases that are identified, including the repressor of silencing 1 (ROS1) [55], Demeter (DME) and Demeter-like 2 and 3 (DML2/3) [56,57] (Figure 1). These four glycosylases can remove 5-mC from any sequence context (mCG, mCHH and mCHG) [58]. ROS1 was the first plant-specific DNA demethylase to be identified. ROS1 demethylates TEs and could influence transposon activity and the transcriptional silencing of nearby genes [59]. ROS1 also induces demethylation in the RdDM-independent regions [60]. DME prefers to be demethylated on the AT-rich transposable elements (TEs), which leads to the expression of the nearby gene changes [61]. The main function of ROS1 is to restrict DNA methylation to its target regions to avoid DNA methylation proliferation and adjacent gene silencing [59]. DME, DML2 and DML3 ensure genomic imprinting in the endosperm, which is essential for seed development [55]. Furthermore, in mammals, 5mC could be actively demethylated through the ten-eleven translocation (TET) dioxygenase-mediated oxidation of 5mC to 5-hydroxymethylcytosine (5hmC), 5-formylcytosine (5fC) and 5-carboxylcytosine (5caC), which is followed by replication-dependent dilution or thymine DNA lycosylase-dependent base excision repair [62].

Reports have characterized the proteins and enzymes of plants that are involved in DNA (de)methylation. However, there is little knowledge about the components controlling targeted DNA (de)methylation during the developmental process [20]. Furthermore, the RdDM model is still not comprehensive; reports showed that RdDM involves allelic interactions. However, these allelic interactions cannot be explained by the existing RdDM model, suggesting that radical changes may be needed in the RdDM model [63]. Additionally, *Arabidopsis thaliana* has been used as a model system to study the basic mechanisms of DNA (de)methylation. One of the reasons for this is that DNA (de)methylation mutants are generally not lethal in *Arabidopsis thaliana* [44]. In recent years, DNA methylation has been found to have regulated many more essential genes for growth and stress responses in plants with more complex genomes, such as rice, maize, tomato and barley [64,65,66], which could reveal new roles for DNA methylation in different plants. 

## 3. Effects of Nutrient Stress on Plant DNA Methylation

Nutrient limitation is major environmental stress that reduces plant growth, productivity and quality [67]. Globally, nitrogen (N) and phosphorus (P) limitations are ubiquitous in soil [68]. Therefore, N and P deficiencies are the main constraints of food production under low-fertilization conditions, while under high-fertilization conditions, large amounts of N and P fertilization can cause large-scale environmental pollution [69]. In addition to N and P, breeding crops with more iron (Fe) and zinc (Zn) is also one of the priorities, since large numbers of people eat grains due to Fe and Zn deficiencies [70,71]. Furthermore, there are essential nutrients for plants, such as sulfur (S), potassium (K), calcium (Ca) and magnesium (Mg) [69,72]. 

DNA methylation in plants plays a vital role in the response to nutrient changes and is involved in controlling nutrient homeostasis [73]. The study of DNA methylation responses to nutrient stress helps breed new nutrient-efficient crops, which help improve food security while reducing environmental impacts [69].

### 3.1. Effects of Nitrogen Stress on Plant DNA Methylation

Nitrogen (N) is one of the crucial macronutrients affecting plant growth and crop yield [74]. When nitrogen is deficient, due to the influence of protein, nucleic acid and phospholipid synthesis in the plant, the plant will grow slowly and dwarf [75]. Epigenetic factors are considered to be among the essential mechanisms for plants in adapting to nitrogen deficiency [76]. Meyer et al. proved that RNA-dependent RNA polymerase2 (RDR2) was involved in the accumulation of biomass under N deficiency in *Arabidopsis thaliana*, which indicated that RdDM could be involved in the regulation of N deficiency [76]. Kou et al. reported that nitrogen deficiency could change DNA methylation in rice. The variation could be inherited by offspring and enhance their tolerance to nitrogen deficiency. Low nitrogen treatment induces the expression of some methylases, such as MET1, DRM1 and DRM2 [64]. Kuhlmann et al. reported that low nitrogen treatment in Arabidopsis thaliana affected eight shoot growth-related SNPs on chromosome 1, resulting in changes in the methylation of their recognition gene regions. They suggested that epigenetic regulation was involved in the nitrogen-use efficiency (NUE) expression of related traits. They also found RdDM-mediated asymmetric cytosine methylation changes, which affected the transcription [77]. Yu et al. reported that nitrogen deficiency resulted in altered methylation patterns in *Leymus chinensis*. They suggested that the cytosine methylation changes around transposable elements were higher than those in other genomic regions [78]. Our previous research reported that the knockdown of the high-affinity nitrate transporter partner protein OsNAR2.1 caused a decrease in nitrogen content in rice and induced DNA methylation reduction [79]. We also found that low nitrogen treatment causes low seed N content, which leads to DNA methylation changes in filial rice [80].

### 3.2. Effects of Phosphorus Stress on Plant DNA Methylation

Phosphorus (P) is an essential macronutrient for plant growth and development [81]. Secco et al. reported that mC changes induced by phosphate starvation occurred preferentially in transposable elements (TEs). They suggested that, during prolonged P deprivation, TEs close to high expression stress-induced genes are hypermethylated without DCL3a, thus preventing their transcription via RNA polymerase II. Furthermore, they found that partial methylation can propagate through mitosis [82]. Yong-Villalobos et al. showed that phosphorus starvation leads to gene-wide methylation changes in *Arabidopsis thaliana*, which are accompanied by changes in gene expression. They found that phosphorus deficiency induced 20% of up-regulated differentially methylated regions (DMRs) in the shoots and 86% of up-regulated DMRs underground. They concluded that DNA methylation changes were required to regulate P sensitive genes, and DNA methylation was necessary for establishing physiological and morphological P starvation responses [83]. Yen et al. showed P deficiency-induced changes in the methylome. They identified over 160 DMRs between low-Pi and Pi-replete conditions. They found that the deubiquitinating enzyme OTU5 is critical for establishing DNA methylation patterns [84]. Tian et al. reported that phosphorus starvation caused an increase in the global methylation level, with millions of differentially methylated cytosines (DmCs) and a few hundred DMRs in tomato. They suggested that methylation changes on P might largely be shaped by TE distributions [65]. Schönberger et al. showed that differential methylation was associated with different P treatments with site-dependent microRNAs (miRNA). Furthermore, some miRNA sequences were directly targeted by differential methylation [85]. Chu et al. reported that low P induced differential methylation, and gene expression showed that the transcriptional alterations of a small part of genes were associated with methylation changes in soybean. They also found that siRNAs modulated TE activity by guiding CHH methylation in TE regions [86].

### 3.3. Effects of Other Nutrient Stresses on Plant DNA Methylation

Zn is an essential micronutrient of all organisms in plants. Mager et al. showed that low Zn treatment could lead to massively reduced DNA methylation, and the enzymes involved in DNA maintenance methylation were repressed. They found that Zn deficiency induced a tremendous reduction in small RNA associated with DNA methylation [87]. Fe is an essential micronutrient in plants. Fe limitation significantly affects plant growth [88]. Sun et al. reported that there is widespread hypermethylation in rice after Fe deficiency, especially in the CHH context. They also found that the transcript abundance of Fe deficiency-induced genes was positive with the 24 nt siRNAs, suggesting that the alteration of methylation patterns is directed by siRNAs, which play an important role in Fe deficiency [88]. Bocchini et al. found that 11 DNA bands were differently methylated in Fe deficiency barleys. Furthermore, their results showed DNA methylation/demethylation patterns very similar to those of barley grown under Fe deprivation in resupplied barley, which indicated that the DNA modifications were heritable [89]. S is an essential element for plant organisms [73]. Huang et al. found that the sulfur accumulation1 (MSA1) mutant *msa1* had a strong S-deficiency response compared with WT. The sulfate transporter genes *SULTR1;1* and *SULTR1;2* were shown to be differentially methylated in *msa1* compared with WT. The results indicated that MSA1 maintained S homeostasis epigenetically via DNA methylation [73]. We summarized the effects of different nutrient stresses on plant methylation in Table 1.

## 4. Methodology of Plant DNA Methylation

DNA methylation research has made significant progress in plants in recent years, and the detection methods have been continuously updated. We summarize the detection methods in plant DNA methylation studies that respond to nutrient stress and other biotic and abiotic stress. The detection of DNA methylation first started in the 1980s. High-performance liquid chromatography (HPLC) was used as the earliest detection method of DNA methylation to measure genomic DNA methylation [90,91]. HPLC is widely used to detect the DNA methylation level of the whole genome of plants, including cotton, tea tree, taxus, etc. [92,93,94,95]. The advantage of this method is that it can measure the DNA methylation level of the whole genome of plants without a reference genome, but the operating system is complicated [96].

Specific-sequence amplified polymorphism (SSAP) and amplified fragment length polymorphism (AFLP) were initially two efficient marker systems for evaluating genetic variation and assessing genetic relationships and were later used for the detection of epigenetic variation [97]. Methylation-sensitive amplified polymorphism (MSAP) is a PCR technology that detects DNA methylation based on amplified fragment length polymorphism (AFLP) technology [98,99]. Reports showed that the epigenetic diversity differed slightly from that of MSAP, AFLP and SSAP [78,97]. As one of the standard methods used in detecting cytosine methylation [64,78,89], the MSAP method uses the restriction enzymes MspI and HpaII to recognize cytosine methylation on the CCGG sequence. The two enzymes have different sensitivities to specific cytosine methylation. HpaII can only recognize mCCGG, the outer methylation site of single-stranded DNA. In contrast, MspI can recognize CmCGG, the inner methylation site of double-stranded or single-stranded DNA. Different bands were amplified from the same DNA sequence to determine the cytosine methylation level at the 5′-CCGG site [100]. The technology is widely used to detect the methylation levels of watermelon, salvia, loquat, poplar, *Viola cazorlensis*, potatoes, cotton, etc. [92,101,102,103]. MSAP technology has high economic efficiency and a low cost. It helps study non-model organisms that lack genome sequencing, and it can screen for mutations and differentiation in the studied genomes [104]. This technology also has certain limitations. Due to the selectivity of the restriction enzymes, some methylation states could be missed [105].

Bisulfite genomic sequencing (BGS) determined the exact positions of 5-methylcytosine on a single strand of DNA [106,107]. By conversing cytosine but not 5mC to uracil, followed by PCR and the sequencing of cloned amplicon DNA, BGS could detect the presence of 5mC at single-nucleotide resolution accurately in a region of interest [106,108]. 

Next-generation sequencing (NGS) technology is the most effective method to identify epigenetic modifications occurring at the DNA level and has been widely used in DNA methylation studies in recent years [109]. Whole-genome bisulfite sequencing (WGBS) technology, also known as MethylC-seq [82,110,111], combines NGS technology with bisulfite conversion methods. It can perform the single-base analysis and genome-wide distribution analysis of DNA methylation in animals and plants [109]. With the development of sequencing technology, WGBS has been performed in many plants, including *Arabidopsis thaliana*, rice, tomato, cucumber and oilseed rape [65,73,77,79,80,83,85,112,113,114]. This method has a high sensitivity to DNA, identifying genome-wide DNA methylation sites with a small number of samples [80]. However, due to its high price, it is primarily use in species with high-quality reference genomes. Compared with WGBS, the reduced representation bisulfite sequencing (RRBS) method is mainly used for the differential analysis of multiple samples with less sequencing. The RRBS method sequences DNA methylation on high-density and representative genes efficiently and accurately. Nevertheless, it is limited by the restriction of enzyme cleavage sites [115]. 

Methylated DNA co-immunoprecipitation sequencing (MeDIP-seq) pre-treats DNA by co-immunoprecipitation and by enriching methylated DNA fragments with anti-methylcytosine nucleoside antibodies. Then, through the high-throughput sequencing of CpG methylated regions, it detects the methylation status and distribution characteristics of the whole genome rapidly and accurately [116,117]. The method has been used in rice, switchgrass, black cottonwood, citrus, etc. [79,118,119,120]. Methyl-CpG-binding domain sequencing (MBD-seq) locates double-stranded methylated DNA fragments using the methyl-binding domain [121]. Both MeDIP and MBD-Seq detected 5mC exclusively, unlike bisulfite conversion, which could not distinguish between 5mC and 5hmC [122]. Moreover, the MeDIP-Seq and Methyl-CpG-binding domain sequencing (MBD-Seq) methods both efficiently detect DNA methylation levels in the whole genome, and their results are generally concordant but non-identical [123,124]. MeDIP-Seq can only find high methylated regions in the genome, such as CpG islands, rather than analyze the single base, and it needs correction with different densities of CpG. MBD-Seq can be separated by different DNA methylations according to the CpG density [125,126].

There are methods that are used less, such as methylation-sensitive single nucleotide primer extension (Ms-SNuPE) [127], methylation-sensitive single-strand conformation analysis (MS-SSCA) [128] and EpiTYPER™ [129]. We list all the methods in Table 2.

## 5. Issues and Prospects

DNA methylation is a reversible epigenetic modification. DNA methylation is involved in multiple cellular and biological processes and plays a critical role in genome stability [24]. In plants, the dynamic regulation of DNA methylation responds to environmental changes and helps plants respond to stress [28,29]. In recent years, there have been many reports about plant DNA methylation, but the signaling and transduction mechanisms involved in DNA methylation are still unclear. Most studies have focused on the DNA methylation expression levels in plants but have focused less on its specific mechanism. It is necessary to conduct further research on how it controls replication initiation. Furthermore, the reports on DNA methylation responses to nutrient stress lack specific sites and specific response mechanisms. Compared with those on 5mC, there are fewer studies on N6-mA in the plant. 6mA DNA methylation is a new epigenetic marker in eukaryotes which has been proven to be a conserved DNA modification that is positively associated with gene expression and contributes to key agronomic traits in plants [130,131]. However, the 6mA changes that respond to nutrient stress remain unclear. Moreover, as we conclude that DNA methylation responds to N, P, Zn, Fe and S, there are rarely reports about DNA methylation responding to other essential elements such as K, Ca, Mg, etc. Therefore, there is still a long way to go in studying the influence of nutrient deficiencies on DNA methylation.

It is necessary to combine DNA methylation modification with histone modification, chromatin remodeling and RNA interference to study the formation and maintenance mechanism of DNA methylation under nutrient stress. This would help to reveal the dynamic changes of methylation during growth and development and to find tissue-specific differences under nutrient stress conditions. The study of DNA methylation responses to nutrient stress could stabilize and improve the yield and quality of crops.

## Figures and Tables

**Figure 1 genes-13-00992-f001:**
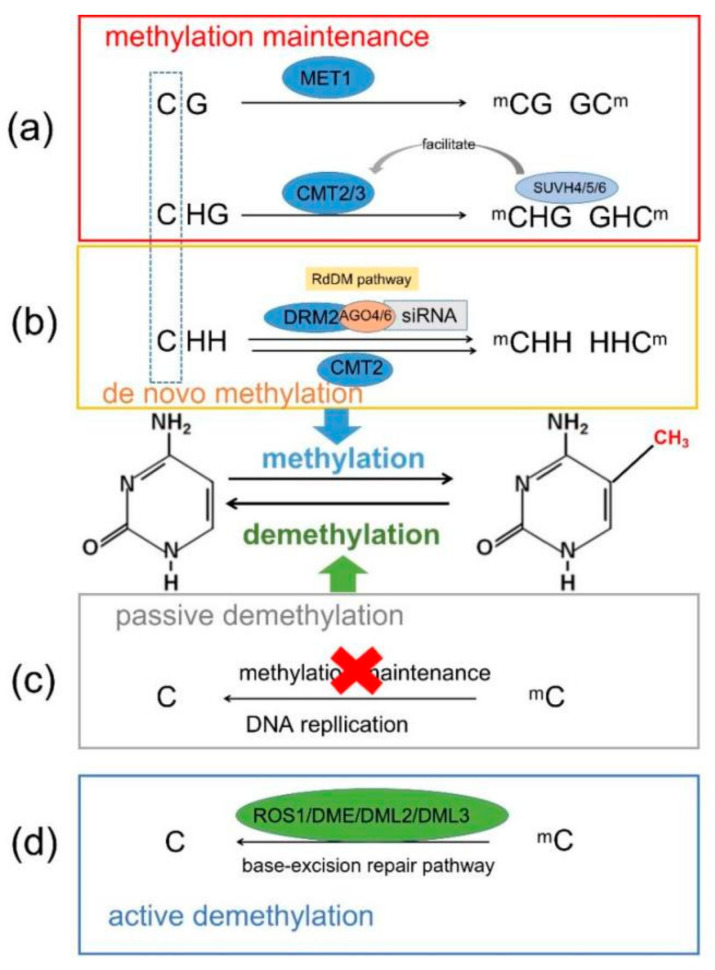
Dynamics of DNA methylation in plants. (H=A, T or C). Two DNA methylation processes in plants: (**a**) DNA methylation maintenance and (**b**) de novo DNA methylation. (**a**) Methyltransferase 1 (MET1) maintains symmetric CG site methylation. Chromosomal methylase (CMT2/3) maintains symmetrical CHG site methylation. The suppressor of the variegation homolog protein, SUVH4, SUVH5 and SUVH6 binds to the methylated CHG site and promotes the function of CMT3/CMT2. (**b**) Asymmetric de novo DNA methylation and CHH methylation performed by domain-rearranged methyltransferase 2 (DRM2) or CMT2, depending on the genomic region. DRM2 causes CHH methylation through the RNA-directed DNA methylation (RdDM) pathway, which depends on the 24 nt small interfering RNA (siRNA). siRNA is loaded onto the ARGONAUTE proteins (AGO), mainly AGO4 and AGO6, interacting with DRM2. DNA demethylation includes (**c**) passive demethylation and (**d**) active demethylation. (**c**) 5mC loses its methyl in passive demethylation during DNA replication. (**d**) 5mC losses are catalyzed by DNA glycosylases in active demethylation. DNA glycosylases including the repressor of silencing 1 (ROS1), Demeter (DME), Demeter-like 2 and 3 (DML2/3).

**Table 1 genes-13-00992-t001:** Summary of the effects of different nutrient stresses on plant methylation.

Element	Plant	Genome Region	Treatment	Mode of Action	Methodology	Reference
N	*Arabidopsis thaliana*	RDR2	−N	RDR2 expression corrlated with morphological traits	Quantitative real-time PCR	[76]
N	*Arabidopsis thaliana*	AT1G55420, AT1G55430 and AT1G55440	−N	DNA methylation change in recognition gene regions (AT1G55420, AT1G55430 and AT1G55440)	WGBS	[77]
N	*Leymus chinensis*	Genomic	−N	Cytosine methylation changes more around transposable elements	AFLP, MSAP, SSAP	[78]
N	Rice	Genomic	−N	Heritable alteration in DNA methylation	MSAP	[64]
N	Rice	Genomic	N content decrease by the knockdown of OsNAR2.1	DNA methylation levels increase in OsNAR2.1 RNAi lines	WGBS, MeDIP	[79]
N	Rice	Genomic	N content decrease in the parent seed	Plant DNA methylation changes induced by parent seed N content	WGBS	[80]
P	Rice	Genomic	−P	DNA methylation occurred preferentially in TEs	MethylC-Seq	[82]
P	*Arabidopsis thaliana*	Genomic	−P	Gene-wide methylation changes	WGBS	[83]
P	*Arabidopsis thaliana*	Genomic	−P	Over 160 DMRs induce by P deficiency	Genome-Wide DNA methylation	[84]
P	Tomato	Genomic	−P	Global methylation level increase	WGBS	[65]
P	*Populus trichocarpa*	Genomic	−P	Differentially methylated miRNAs	WGBS	[85]
P	Soybean	Genomic	−P	Differential methylation, and siRNAs modulated TE activity by guiding CHH methylation	BGS	[86]
Zn	Maize	Genomic	−Zn	Major methylation loss, mostly in transposable elements	BGS	[87]
Fe	Rice	Genomic	−Fe	Hypermethylation, especially for the CHH	MethylC-Seq	[88]
Fe	Barley	Genomic	−Fe	Eleven DNA bands differently methylated the	MSAP	[89]
S	*Arabidopsis thaliana*	SULTR1.1 and SULTR1.2	−S	DNA methylation of SULTR1.1 and SULTR1.2 changes in *msa1*	WGBS	[73]

Note: −N: nitrogen deficiency; −P: phosphorus deficiency; −Zn: zinc deficiency; −Fe: iron deficiency; −S: sulfur deficiency; WGBS/MethylC-Seq: Whole-genome bisulfite sequencing; AFLP: Amplified fragment length polymorphism; MSAP: Methylation-sensitive amplified polymorphism; SSAP: Specific-sequence amplified polymorphism; MeDIP: Methylated DNA co-immunoprecipitation sequencing; BGS: Bisulfite genomic sequencing.

**Table 2 genes-13-00992-t002:** Methodology of plant DNA methylation.

Methods	Coverage	Reference Genome	Advantage	Limitation	Reference
HPLC	Genomic DNA	No	Do not need a reference genome	Complicated operating system	[91]
SSAP	CG region	No	High economic efficiency without a reference genome	Not specifically designed to detect methylation	[97]
AFLP	CG region	No	High economic efficiency without a reference genome	Not specifically designed to detect methylation	[97]
MSAP	CG region	No	High economic efficiency without a reference genome	Miss methylation states	[99]
BGS	Genomic DNA	Yes	Detects the presence of 5mC at the single-nucleotide resolution accurately	Only in the specific region	[106]
WGBS/ MethylC-Seq	Genomic DNA	Yes	High sensitivity to DNA	High price	[109]
RRBS	Promoters and CpG islands	Yes	Efficient and accurate on the high-density and representative genes	Limited by enzyme cleavage sites	[115]
MeDIP-Seq	CG region	Yes	Detects the CpG island of the whole genome rapidly and accurately	Cannot analyze the single base and needs correction with different densities of CpG	[116]
MBD-Seq	CG region	Yes	Separated different DNA methylation according to CpG density	Antibodies may cross-react	[126]
MS-SSCA	Individual CpG site	No	Fast	Primer design is complex	[128]
Ms-SNuPE	CG region	No	Analysis of C and T content representing the degree of DNA methylation	The number of each analysis is small	[127]
EpiTYPER™	CG region	No	Fast and reproducible	DNA methylation status is unclear, with overlapping CpGs	[129]

Note: HPLC: High-performance liquid chromatography; SSAP: Specific-sequence amplified polymorphism; AFLP: Amplified fragment length polymorphism; MSAP: Methylation-sensitive amplified polymorphism; BGS: Bisulfite genomic sequencing; WGBS/MethylC-Seq: Whole-genome bisulfite sequencing; RRBS: Reduced representation bisulfite sequencing; MeDIP: Methylated DNA co-immunoprecipitation sequencing; MBD-Seq: Methyl-CpG-binding domain sequencing; MS-SSCA: Methylation-sensitive single-strand conformation analysis; Ms-SNuPE: Methylation-sensitive single nucleotide primer extension.

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
