# Peer review of "Plant DNA Methylation Responds to Nutrient Stress"

_genes, 2022, doi:10.3390/genes13060992_

Round 1
Reviewer 1 Report
Review: Research Advances on Plant DNA methylation under nutrient stress
2022
This manuscript aims to review updated knowledge regarding the connection between DNA methylation and nutrient deficiency which is indeed a very interesting and innovative point for enhanced plant protection and stress tolerance.
Regarding the section on plant methylation patterns, many reviews have been written on this subject and this MS review does not add novel information. In this section, some sentences are vague and would need to be reformulated giving more details to be more robust and to facilitate the reading and comprehension of readers. The scheme of figure 1 could include more updated information in order to illustrate how complex DNA methylation dynamics is and what is still not known. These comments also apply to section 4 where the techniques available to study plant DNA methylation are presented.
Regarding the section 3 dedicated to literature evidences for the connection between DNA methylation flexibility and nutrients availability, more information and in particular a deep mechanistic interpretation of the literature could be included to enrich this interesting point in the MS.
Author Response
This manuscript aims to review updated knowledge regarding the connection between DNA methylation and nutrient deficiency which is indeed a very interesting and innovative point for enhanced plant protection and stress tolerance.
Q1:Regarding the section on plant methylation patterns, many reviews have been written on this subject and this MS review does not add novel information. In this section, some sentences are vague and would need to be reformulated giving more details to be more robust and to facilitate the reading and comprehension of readers.
A1: Thank you for your comments. We accept review’s suggestion. We add more detail in “2. Plant DNA methylation patterns” from line 75 to 113, “The suppressor of variegation homolog protein, SUVH4, SUVH5, and SUVH6, binds to the methylated CHG site and promotes the function of CMT3/CMT2 [37,38]. While asymmetric de novo DNA methylation, CHH methylation is performed by domain-rearranged methyltransferase 2 (DRM2) or CMT2, depending on the genomic region. DRM2 causes CHH methylation through a plant-specific mechanism, the RNA-directed DNA methylation (RdDM) pathway, which depends on the 24nt small interfering RNA (siRNA) [30,39,40].
Most of the RdDM pathway researches are studied in Arabidopsis [30,40,41]. RNA polymerase IV (Pol IV), as the template for RNA-dependent RNA polymerase 2 (RDR2), mediated the generation of double-stranded RNAs (dsRNA)[39,40]. Then the DICER-like protein (DCL2/3/4) cleaves the dsRNAs to generate 24 nt siRNAs. siRNA is loaded onto the ARGONAUTE proteins (AGO), mainly AGO4 and AGO6, which interacts with DRM2 to catalyze de novo DNA methylation [30,33,42] (Fig.1). This process is possible assisted by RNA-DIRECTED DNA METHYLATION 1 (RDM1), which may bind single-stranded methylated DNA [43]. While at some RdDM loci, Pol II produces 24 nt siRNAs and scaffold RNAs [44]. Furthermore, at some transposons, POL II and RDR6 produce precursors of 21nt or 22nt siRNAs [45-47].
On the other hand, replacing 5mC with unmethylated cytosine is equally important in regulating gene expression as cytosine methylated. DNA demethylation includes passive demethylation and active demethylation [48,49]. In passive demethylation, 5mC loses its methyl during DNA replication, and in active demethylation, 5mC loses catalyzed by DNA glycosylases [50]. Passive demethylation is a nuclear factor (NF) that adheres to the 5mC during DNA replication and blocks the maintenance of DNA methylation, which leads to the loss of DNA methylation in the newly synthesized strand [48,50]. Active demethylation balances the methylation level of the genome and maintains gene expression. The excision of C5-methylcytosine achieves the active demethylation of 5mC by the DNA glycosylase, and then repairs cytosine by the base excision through the base-excision repair (BER) pathway [34]. There are four DNA glycosylases are identified, including repressor of silencing 1 (ROS1) [51], Demeter (DME), Demeter-like 2 and 3 (DML2/3) [52,53] (Fig.1). These four glycosylases can remove 5-mC from any sequence context (mCG, mCHH and mCHG) [54]. ROS1 was the first identified plant-specific DNA demethylase. ROS1 demethylates TEs and could influence transposon activity and transcriptional silencing of nearby genes [55]. ROS1 also induce demethylation in the RdDM-independent regions[56]. DME prefers demethylated on the AT-rich transposable elements (TEs), which leads to the expression of the nearby genes changes[57]. The main function of ROS1 is to restrict DNA methylation to its target regions to avoid DNA methylation proliferation and adjacent gene silencing [55]. DME, DML2 and DML3 ensure genomic imprinting in the endosperm, which is essential for seed development [51].”
Q2: The scheme of figure 1 could include more updated information in order to illustrate how complex DNA methylation dynamics is and what is still not known.
A2: We appreciate reviewer’s comments.
(1) We adjust our Figure 1 according to reviewer’s suggestion.
Figure 1. Dynamics of DNA methylation in plants. (H=A,T or C). Two DNA methylation processes in plants: (a) DNA methylation maintenance and (b) de novo DNA methylation. (a) Methyltransferase 1 (MET1) maintains symmetric CG site methylation. Chromosomal methylase (CMT2/3) maintains symmetrical CHG site methylation. The suppressor of variegation homolog protein, SUVH4, SUVH5, and SUVH6, binds to the methylated CHG site and promotes the function of CMT3/CMT2. (b) Asymmetric de novo DNA methylation, CHH methylation, performed by domain-rearranged methyltransferase 2 (DRM2) or CMT2 depending on the genomic region. DRM2 causes CHH methylation through the RNA-directed DNA methylation (RdDM) pathway, which depends on the 24nt small interfering RNA (siRNA). siRNA is loaded onto the ARGONAUTE proteins (AGO), mainly AGO4 and AGO6, interacting with DRM2. DNA demethylation includes (c) passive demethylation and (d) active demethylation. (c) 5mC loses its methyl in passive demethylation during DNA replication. (d) 5mC loses catalyzed by DNA glycosylases in active demethylation. DNA glycosylases including repressor of silencing 1 (ROS1), Demeter (DME), Demeter-like 2 and 3 (DML2/3)
(2) We add discussion from 129 to 141 to explain what is still not known, “Reports have characterized plants' proteins and enzymes involved in DNA (de)methylation. However, there is little knowledge about the components controlling targeted DNA (de)methylation during the developmental process [40]. Furthermore, the RdDM model is still not comprehensive, reports showed that RdDM involves allelic interactions. However, these allelic interactions cannot be explained by the existing RdDM model, suggesting that radical changes may be needed in the RdDM model [58]. Also, Arabidopsis thaliana has been used as a model system to study the basic mechanisms of DNA (de)methylation. One of the reasons is that DNA (de)methylation mutants are generally not lethal in Arabidopsis thaliana [39]. In recent years, DNA methylation has been found regulated many more essential genes for growth and stress responses in plants with more complex genomes, like rice, maize, tomato and barley [59-61], which could reveal new roles for DNA methylation in different plants. ”.
Q3: These comments also apply to section 4 where the techniques available to study plant DNA methylation are presented.
A3: We appreciate reviewer’s comments. And we accepted the suggestion.
We replenish more techniques in “4. Methodology of plant DNA methylation”
Line 242-249: “Specific-sequence amplified polymorphism (SSAP) and amplified fragment length polymorphism (AFLP) are two efficient marker systems for evaluating genetic variation and assessing genetic relationships initially and were used for the detection of epigenetic variation later [92]. Methylation-sensitive amplified polymorphism (MSAP) is a PCR technology that detects DNA methylation-based on amplified fragment length polymorphism (AFLP) technology [93,94]. Reports showed that the epigenetic diversity differed slightly from MSAP, AFLP and SSAP [73,92]. ”
Line 262-267: “Bisulfite genomic sequencing (BGS) determined the exact positions of 5-methylcytosine on a single strand of DNA [101,102]. By conversing cytosine but not 5mC to uracil, followed by PCR and sequencing of cloned amplicon DNA. BGS could detect the presence of 5mC at single-nucleotide resolution accurately in a region of interest [101,103]. ”
And we adjusted Table 2 according to the section.
Table 2. Methodology of plant DNA methylation.
|
Methods |
Coverage |
Reference genome |
Advantage |
Limitation |
Reference |
|
HPLC |
Genomic DNA |
No |
Don't need a reference genome |
Complicate operating system |
Wanger et al, 1981 |
|
SSAP |
CG region |
No |
High economic efficiency without a reference genome |
Not specifically designed to detect methylation |
Shan et al. 2012 |
|
AFLP |
CG region |
No |
High economic efficiency without a reference genome |
Not specifically designed to detect methylation |
Shan et al. 2012 |
|
MSAP |
CG region |
No |
High economic efficiency without a reference genome |
Miss methylation states |
Xiong et al., 1999 |
|
BGS |
Genomic DNA |
Yes |
Detect the presence of 5mC at single-nucleotide resolution accurately |
Only in the specific region |
Darst et al. 2010 |
|
WGBS/ MethylC-Seq |
Genomic DNA |
Yes |
High sensitivity to DNA |
High price |
Baubec and Akalin, 2016 |
|
RRBS |
Promoters and CpG islands |
Yes |
Efficient and accurate on the high-density and representative genes |
Limited by enzyme cleavage sites |
Schmidt et al., 2017 |
|
MeDIP-Seq |
CG region |
Yes |
Detect CpG island of the whole genome rapidly and accurately |
Can't analyze the single base, and needs correction with different density of CpG |
Seymour et al., 2017 |
|
MBD-Seq |
CG region |
Yes |
Separated different DNA methylation according to cpg density |
Antibodies may cross-react |
Ahn et al., 2021; Nair et al., 2014 |
|
MS-SSCA |
Individual CpG site |
No |
Fast |
Primer design is complex |
Rodríguez López et al., 2010 |
|
Ms-SNuPE |
CG region |
No |
Analysis of C and T content representing the degree of DNA methylation |
The number of each analysis is small |
Gonzalgo and Liang, 2007 |
|
EpiTYPER™ |
CG region |
No |
Fast and reproducible |
DNA methylation status is unclear with overlapping CpGs |
Kunze, 2018 |
Note: HPLC: High-performance liquid chromatography; SSAP: Specific-sequence amplified polymorphism; AFLP: Amplified fragment length polymorphism; MSAP:Methylation-sensitive amplified polymorphism; BGS: Bisulfite genomic sequencing; WGBS/MethylC-Seq: Whole-genome bisulfite sequencing; RRBS: Reduced representation bisulfite sequencing; MeDIP:Methylated DNA co-immunoprecipitation sequencing; MBD-Seq: Methyl-CpG-binding domain sequencing; MS-SSCA:Methylation‑sensitive single‑strand conformation analysis; Ms-SNuPE: Methylation‑sensitive single nucleotide primer extension.
Q4: Regarding the section 3 dedicated to literature evidences for the connection between DNA methylation flexibility and nutrients availability, more information and in particular a deep mechanistic interpretation of the literature could be included to enrich this interesting point in the MS.
A4: We appreciate reviewer’s comments. And we accepted the suggestion.
We add more information and more deep mechanistic interpretation in section “3. Effects of nutrient stress on plant DNA methylation”
Line 160-164: “Meyer et al. proved that RNA-dependent RNA polymerase2 (RDR2) was involved in the accumulation of biomass under N deficiency in Arabidopsis thaliana, which indicated that RdDM could be involved in the regulation of N deficiency [71].”
Line 167-172: “Kuhlmann et al. reported that low nitrogen treatment in Arabidopsis affected eight shoot growth-related SNPs on chromosome 1, resulting in changes in the methylation of their recognition gene regions. They suggested that epigenetic regulation was involved in nitrogen-use efficiency (NUE) expression of related traits. They also found RdDM-mediated asymmetric cytosine methylation changes, which affected the transcription [72]”
Line 182-186: “Secco et al. reported that mC changes induced by phosphate starvation occurred preferentially in transposable elements (TEs). They suggested that during the prolonged P deprivation, TEs close to high expression stress-induced genes are hypermethylated without DCL3a, thus preventing their transcription via RNA polymerase II. Furthermore, they found partial methylation can propagate through mitosis.[77].”
Line 186-193: “Yong-Villalobos et al. showed that phosphorus starvation leads to gene-wide methylation changes in Arabidopsis thaliana, accompanied by changes in gene expression,. They found phosphorus deficiency induced 20%up-regulation differentially methylated regions (DMRs) in the shoots and 86% DMR up-regulated underground. They concluded that DNA methylation changes were required to regulate of P sensitive genes, and DNA methylation was necessary for establishing physiological and morphological P starvation responses [78].”
Line 196-202: “DNA methylation patterns [79]. Tian et al. reported phosphorus starvation caused an increase in the global methylation level, with millions of differentially methylated cytosines (DmCs) and a few hundred DMRs in tomato. They suggested methylation changes on P might largely be shaped by TE distributions [60]. Schönberger et al. showed that differential methylation was associated with different P treatments with site-dependent microRNAs. Furthermore, some miRNAs sequences were directly targeted by differential methylation [80]. “

Reviewer 2 Report
In this manuscript (genes-1722133) entitled "Research advances on plant DNA methylation under nutrient stress" submitted to Genes, authors reviewed recent developments in the plant DNA methylation patterns, the effects of nutrient stress, like N, P, Fe, Zn and S stress, on plant DNA methylation and research techniques of plant DNA methylation,which would certainly facilitate the future research in the investigation of DNA methylation and plant nutrient stress. The writing is clear and concise and in good English.
No major issues are of concern, however some points or minor issues needs addressing to improve the quality of this manuscript.
- Section 4: Research techniques of plant DNA methylation is not linked to plant nutrition research, please rewrite this section and connect with plant nutrition research.
- Line 68: DNMTs do not involve in DNA demethylation, please revise.
- Line 106: the comma in 'sulphur, potassium' is incorrect,, please revise.
- Arabidopsis thaliana and Leymus chinensis shoule be in italics, please revise.
- Full names of the abbreviations N, P, Fe, Zn, S, and NUE should be spelt out at their first appearance in this article.
Author Response
In this manuscript (genes-1722133) entitled "Research advances on plant DNA methylation under nutrient stress" submitted to Genes, authors reviewed recent developments in the plant DNA methylation patterns, the effects of nutrient stress, like N, P, Fe, Zn and S stress, on plant DNA methylation and research techniques of plant DNA methylation,which would certainly facilitate the future research in the investigation of DNA methylation and plant nutrient stress. The writing is clear and concise and in good English.
No major issues are of concern, however some points or minor issues needs addressing to improve the quality of this manuscript.
Q1:Section 4: Research techniques of plant DNA methylation is not linked to plant nutrition research, please rewrite this section and connect with plant nutrition research.
A1: We appreciate reviewer’s comments. And we accepted the suggestion.
We add methodology in table 1 to link research techniques of plant DNA methylation to plant nutrition research. And we add in line 223-225 “We summarize the detection methods in plant DNA methylation studies that respond to nutrient stress and other biotic and abiotic stress. ”
Table 1. Summary of effects of different nutrients tress on plant methylation.
|
Element |
Plant |
Genome region |
Treatment |
Mode of action |
Methodology |
Reference |
|
N |
Arabidopsis thaliana |
RDR2 |
-N |
RDR2 expression corrlated with morphological traits |
Quantitative real-time PCR |
Meyer et al.2019 |
|
N |
Arabidopsis thaliana |
AT1G55420, AT1G55430 and AT1G55440 |
-N |
DNA methylation change in recognition gene regions (AT1G55420, AT1G55430 and AT1G55440) |
WGBS |
Kuhlmann et al., 2020 |
|
N |
Leymus chinensis |
Genomic |
-N |
Cytosine methylation changes more around transposable elements |
AFLP, MSAP, SSAP |
Yu et al., 2013 |
|
N |
Rice |
Genomic |
-N |
Heritable alteration in DNA methylation |
MSAP |
Kou et al., 2011 |
|
N |
Rice |
Genomic |
N content decrease by knockdown of OsNAR2.1 |
DNA methylation levels increase in OsNAR2.1 RNAi lines |
WGBS, MeDIP |
Fan et al. 2020 |
|
N |
Rice |
Genomic |
N content decrease in parent seed |
Plant DNA methylation changes induce by to parent seed N content |
WGBS |
Fan et al. 2021 |
|
P |
Rice |
Genomic |
-P |
DNA methylation occurred preferentially in TEs |
MethylC-Seq |
(Secco et al., 2015 |
|
P |
Arabidopsis thaliana |
Genomic |
-P |
Gene-wide methylation changes |
WGBS |
Yong-Villalobos et al., 2015 |
|
P |
Arabidopsis thaliana |
Genomic |
-P |
Over 160 DMRs induce by P deficiency |
Genome-Wide DNA methylation |
Yen et al., 2017 |
|
P |
Tomato |
Genomic |
-P |
Global methylation level increase |
WGBS |
Tian et al., 2021 |
|
P |
Populus trichocarpa |
Genomic |
-P |
Differentially methylated miRNAs |
WGBS |
Schönberger et al., 2016 |
|
P |
Soybean |
Genomic |
-P |
Differential methylation, and siRNAs modulated TE activity by guiding CHH methylation |
BGS |
Chu et al., 2020 |
|
Zn |
Maize |
Genomic |
-Zn |
Major methylation loss, mostly in transposable elements |
BGS |
Mager et al., 2018 |
|
Fe |
Rice |
Genomic |
-Fe |
hypermethylation, especially for the CHH |
MethylC-Seq |
Sun et al., 2021 |
|
Fe |
Barley |
Genomic |
-Fe |
Eleven DNA bands differently methylated |
MSAP |
Bocchini et al., 2015 |
|
S |
Arabidopsis thaliana |
SULTR1.1 and SULTR1.2 |
-S |
DNA methylation of SULTR1.1 and SULTR1.2 changes in msa1 |
WGBS |
Huang et al., 2016 |
Note: -N: nitrogen deficiency; -P: phosphorus deficiency; -Zn: zinc deficiency; -Fe: iron deficiency; -S: sulfur deficiency. WGBS/MethylC-Seq: Whole-genome bisulfite sequencing; AFLP: Amplified fragment length polymorphism; MSAP:Methylation-sensitive amplified polymorphism; SSAP: Specific-sequence amplified polymorphism; MeDIP:Methylated DNA co-immunoprecipitation sequencing; BGS: Bisulfite genomic sequencing
Q2: Line 68: DNMTs do not involve in DNA demethylation, please revise.
A2: We appreciate reviewer’s comments. And we revised and corrected the mistake in line 66-68
“Different DNMTs involves in two DNA methylation processes in plants: DNA methylation maintenance and de novo DNA methylation (Fig.1).”
Q3: Line 106: the comma in 'sulphur, potassium' is incorrect,, please revise.
A3: We appreciate reviewer’s comments. And we revised and corrected the mistake in line 150-151
“Furthermore, there are essential nutrients for plants, like sulphur (S), potassium (K), calcium (Ca) and magnesium (Mg) [64,67]”
Q4: Arabidopsis thaliana and Leymus chinensis shoule be in italics, please revise.
A4: We appreciate reviewer’s comments. And we revised and corrected the mistake in Table1, and we checked the whole manuscript to make sure there is no similar mistake.
Q5: Full names of the abbreviations N, P, Fe, Zn, S, and NUE should be spelt out at their first appearance in this article.
A5: We appreciate reviewer’s comments. And we revised and corrected the mistake. We checked all the abbreviations their first appearance.
Like in line 144-145: “Globally, nitrogen (N) and phosphorus (P) limitations are ubiquitous in soil [63]. ”
Line 148-151: “In addition to N and P, breeding crops with more iron (Fe) and zinc (Zn) is also one of the priorities since large numbers of people eat grains due to Fe and Zn deficiencies [65,66]. Furthermore, there are essential nutrients for plants, like sulphur (S), potassium (K), calcium (Ca) and magnesium (Mg) [64,67].”
Line 170-171: “They suggested that epigenetic regulation was involved in nitrogen-use efficiency (NUE) expression of related traits.”

Round 2
Reviewer 1 Report
Review: Research Advances on Plant DNA methylation under nutrient stress
Revised review: This manuscript has been reviewed by the authors and it certainly improved a lot the message, nevertheless, it still needs some updates namely at the following points:
Lines 32-43: It needs a final sentence with a critical view of what is still not understood/known, for example, how many generations are needed to establish an epigenetic memory?
Lines 52-53: The following sentence needs clarification: “the reversibility of DNA methylation avoids excessive gene recombination and population diversity”.
Line 104: The vision of methylation/demethylation cycle still needs to be updated. At least regarding active DNA demethylation, the process has been a matter of debate with the involvement of many enzymes. Here, for example there is no reference to the role of TET enzymes on the dynamics of this process. Certainly, the sentence “in active demethylation, 5mC loses catalyzed by DNA glycosylases” it needs reformulation.
Line 308-309: “compared between MeDIP-Seq and Methyl-CpG-binding domain sequencing (MBD-Seq), they both have high efficiently detection of DNA methylation levels in whole genome” this sentence needs clarification since MeDIP seq only considers the immunoprecipitated fraction of DNA.
Line 344: “DNA methylation analyzes the complex interaction between genotype and environment” reformulate/clarify this sentence.
Author Response
Thank you very much for reviewing our manuscript. We have substantially revised our manuscript after reading your comments.
Revised review: This manuscript has been reviewed by the authors and it certainly improved a lot the message, nevertheless, it still needs some updates namely at the following points:
Q1: Lines 32-43: It needs a final sentence with a critical view of what is still not understood/known, for example, how many generations are needed to establish an epigenetic memory?
A1: Thank you for the comments and we accepted the reviewer’s suggestion.
We added discussion and a final sentence with a critical view of what is still not understood/known in Lines 43-51:”It is already proven that the stress-induced changes in DNA methylation could be partially inherited by the next generation, which preferentially occurs through the female germ line [18,19]. Such heredity was being considered a source of diversity, which could be utilized in breeding programs [20]. Therefore, the study of plant epigenetic mechanisms has great significance for crop cultivation [21]. However, persistent stress is vital for establishing DNA methylation-dependent stress memory in plants [22]. If the progeny were not continuously stressed, the inherited epigenetic status is gradually reset[19]. But how many generations are needed to establish an epigenetic memory is still unclear.”
Q2: Lines 52-53: The following sentence needs clarification: “the reversibility of DNA methylation avoids excessive gene recombination and population diversity”.
A2: Thank you for the comments and we accepted the reviewer’s suggestion.
We clarified the sentence in Lines 52-53 to Lines 59-61: “The reversibility of DNA methylation could rapidly and reversibly modify plant genomic DNA, which avoids excessive gene recombination and population diversity [30]”
Q3: Line 104: The vision of methylation/demethylation cycle still needs to be updated. At least regarding active DNA demethylation, the process has been a matter of debate with the involvement of many enzymes. Here, for example there is no reference to the role of TET enzymes on the dynamics of this process. Certainly, the sentence “in active demethylation, 5mC loses catalyzed by DNA glycosylases” it needs reformulation.
A3: Thank you for the comments. Our review focused on plant methylation. Therefore we didn't involve TET enzymes, which regulated mammals' demethylation. But we accepted the review's comment and made revisions based on the reviewer's comments.
Lines 124-129 “Furthermore, in mammals, 5mC could be active demethylated through ten-eleven translocation (TET) dioxygenase-mediated oxidation of 5mC to 5-hydroxymethylcytosine (5hmC), 5-formylcytosine (5fC) and 5-carboxylcytosine (5caC), which followed by replication-dependent dilution or thymine DNA lycosylase dependent base excision repair [62].”
Q4: Line 308-309: “compared between MeDIP-Seq and Methyl-CpG-binding domain sequencing (MBD-Seq), they both have high efficiently detection of DNA methylation levels in whole genome” this sentence needs clarification since MeDIP seq only considers the immunoprecipitated fraction of DNA.
A4: Thank you for the comments, and we accepted the reviewer’s suggestion. We made revisions about MeDIP and MBD-Seq based on review’s comment.
Lines 303-309 “Methyl-CpG-binding domain sequencing (MBD-seq) located double-stranded methylated DNA fragments using the methyl-binding domain [121]. Both MeDIP and MBD-Seq detected 5mC exclusively, unlike bisulfite conversion, which couldn’t distinguish between 5mC and 5hmC [122]. Moreover, compared between MeDIP-Seq and Methyl-CpG-binding domain sequencing (MBD-Seq), they both efficiently detect DNA methylation levels in the whole genome, and their results are generally concordant but non-identical [123,124].”
Q5: Line 344: “DNA methylation analyzes the complex interaction between genotype and environment” reformulate/clarify this sentence
A5: Thank you for the comments, and we accepted the reviewer’s suggestion. Because this sentence was ambiguous, we deleted this sentence in Line 342.
